# Study on Coated Zr-V-Cr Getter with Pore Gradient Structure for Hydrogen Masers

**DOI:** 10.3390/ma15176147

**Published:** 2022-09-05

**Authors:** Jiale Zhang, Huihui Song, Jinyu Fang, Xueling Hou, Shuiming Huang, Jie Xiang, Tao Lu, Chao Zhou

**Affiliations:** 1School of Materials Science and Engineering, Shanghai University, Shanghai 200072, China; 2Shanghai Kingv Material Technology Co., Ltd., Shanghai 200000, China; 3National Key Laboratory of Science and Technology on Vacuum Technology and Physics, Lanzhou Institute of Physics, Shanghai 201204, China

**Keywords:** hydrogen maser, Zr-V-Cr getter, pore gradient structure, random vibration test, thermal infrared detection

## Abstract

As the core component of satellite navigation, the hydrogen maser needs a high vacuum environment to maintain the stability of the frequency signal. The getter pump, composed of various non-evaporable getters, plays an important role in maintaining the high vacuum. In this paper, the Zr_100-x_Cu_x_ (x = 0, 2, 4, 6)/Zr_56.97_V_35.85_Cr_7.18_ getter was studied and the contradiction between sorption performance and mechanical properties was solved. The Zr-V-Cr getter, a better candidate for getter pump, exists for problems which will destroy the high vacuum and affect the service life of the hydrogen maser. To solve the problem of dropping powder from Zr-V-Cr getter, Zr-Cu films were coated on the surface of Zr-V-Cr matrix to obtain the pore gradient structure. After vacuum sintering, the interface showed gradient structure and network change in pore structure from Zr-Cu film to Zr-V-Cr matrix. These characteristic structures made Zr-V-Cr getter have good absorption properties, which is better than a similar product of SAES company and mechanical properties. Because the Zr-Cu film on Zr-V-Cr matrix effectively prevented dropping powders from the matrix, (Zr-Cu)/(Zr-V-Cr) getter solved the problem of dropping powder. The self-developed new getter with pore gradient structure is of great significance for maintaining the high vacuum of hydrogen maser in the future.

## 1. Introduction

Hydrogen maser, known as the heart of satellite, is the core component of satellite navigation. Its high-precision timing and time-frequency standard can provide effective “time scale/positioning” support for satellite navigation system [1,2,3]. For example, China’s BeiDou Satellite System is equipped with hydrogen masers [4,5]. The hydrogen maser has a high-vacuum during service. If the system vacuum reduces, the collision probability between the hydrogen atom in the atomic storage bubble and other particles will increase, resulting in the shortening of the interaction time between the hydrogen maser of the effective energy level and the resonant cavity. It is very important to maintain a high atomic vacuum in hydrogen masers. The getter pump [6] plays an important role in maintaining the high vacuum of the hydrogen maser. It can absorb the gases affecting the vacuum, such as low-energy H_2_, CO, O_2_, H_2_O, and CO_2_, in the service environment. The reason for the suction of the getter pump is that the getters constitute the getter pump play roles [7]. Therefore, the quality of the getter has great impacts on the vacuum system of the hydrogen maser.

Titanium porous materials are the main getters for adsorption pumps of hydrogen maser in China [8,9,10,11]. However, Ti-based getters have the problem of high activation temperature (700~900 °C) [12]. When designing the heating system of getter pump, the requirements of high-temperature mechanical properties and high-temperature deformation resistance of its supporting materials are put forward. At the same time, high temperature heating will cause thermal damage to vacuum devices. In addition, the solubility of hydrogen in titanium is limited. For example, α-Ti has the best hydrogen absorption capacity at 300 °C, but it only reaches about 7% (atomic fraction). When the temperature drops to room temperature, the hydrogen absorption capacity will be further weakened, and the maximum hydrogen content in α-Ti is about 0.04% [13]. Excessive hydrogen absorption of getters will lead to embrittlement.

In addition to Ti-based getters, Zr-based getters [14,15,16,17] are also widely used in high vacuum systems. Zr-based getters have good gettering performance, especially Zr-V-Fe getter. Jin YF et al. [18] found that the hydrogen absorption capacity of Zr-V-Fe alloy was increased and the hysteresis coefficient improved due to the increase of V element. The comprehensive performance was excellent. However, getters have the possibility of powder loss in application. Many scholars have carried out relevant research on the prevention of powder loss. For example, Zhang Yan [19] et al. studied Ti-Mo material as matrix and added Zr-V-Fe alloy to form composite material, but the problem of powder removal had not been satisfactorily solved. Once the particles of getters fall off, it will destroy the high vacuum system and seriously affect the service life of the hydrogen maser.

Compared with Ti-based getters, Zr-based getters, and Zr-V-Fe getter, Zr-V-Cr getter has lower activation temperature (500 °C) [20]. The maximum hydrogen absorption capacity at room temperature after activation is 2.25 wt.% [21]. However, Zr-V-Cr getter also has the contradiction between gettering performance and mechanical properties as other getters. Therefore, there is also the possibility of powder loss in application.

On the basis of previous studies, Zr_56.97_V_35.85_Cr_7.18_ getter with the best performance is selected as the matrix [14], the Zr-Cu film is coated on the Zr-V-Cr matrix, and then sintered to obtain the gradient change of pore structure [22] from the matrix to the film. The film is firmly bonded to the matrix, so the getter has good mechanical properties and anti-dropping powder ability while maintaining high sorption characteristics. The structural schematic diagram is shown in Figure 1. At the same time, the mechanical properties of microelectronic devices of (Zr-Cu)/(Zr-V-Cr) getter is tested to ensure practical application in the hydrogen maser.

In this paper, the preparation and properties of gradient structure (Zr-Cu)/(Zr-V-Cr) getter are studied in order to solve the contradiction between strength and gettering performance and the problem of dropping powders in applications of Zr-V-Cr getter.

## 2. Materials and Methods

(1)Preparation of Zr-V-Cr matrix samples:

The raw materials for the matrix of this experiment were Zr (purity not less than 99.95%), V (purity not less than 99.99%), and Cr (purity not less than 99.95%). The raw materials used for the matrix are all from the manufacturer Zhongnuo new materials (Beijing, China) Technology Co., Ltd. Considering the V element is volatile in the melting process, 2% more raw materials were added on the basis of V mass. The Zr_56.97_V_35.85_Cr_7.18_ alloy as the matrix of (Zr-Cu)/(Zr-V-Cr) getter was melted in a non-consumable vacuum arc melting furnace. During smelting, the electric arc furnace was 150 A, and the smelting time was 10 s each time. The ingots were re-melted four times for homogenization. Then we crushed the ingots, milling by wet ball milling under a protective atmosphere to prepare alloy powder having 300 mesh. Placing the obtained powder into the mold, a disc with an inner diameter of 4 mm and an outer diameter of 10 mm and pressing. Then we sintered the prepared samples under 3.0 × 10^−3^ Pa, sintering at 850 °C for 60 min. Zr-V-Cr matrix samples were prepared.

(2)Preparation of Zr_100-x_Cu_x_ (x = 0, 2, 4, 6) film slurry:

The raw materials for the film of this experiment were Zr (purity not less than 99.95%) and Cu (purity not less than 99.9%). The raw material powder used for the film layer is from Hunan Zhuzhou Runfeng Co., Ltd. (Zhuzhou, China). The Zr_100-x_Cu_x_ (x = 0, 2, 4, 6) slurry as the film of (Zr-Cu)/(Zr-V-Cr) getter was prepared by 400 mesh Zr powders and 400 mesh Cu powders according to the mass ratio of Zr_100-x_Cu_x_ (x = 0, 2, 4, 6). Next, we added 20 mL of alcohol into the ball mill. During wet ball milling, rotate clockwise for 30 min and then counterclockwise for 30 min. The number of coated slurries was as follows:(3)Preparation of (Zr_100-x_-Cu_x_)/(Zr-V-Cr)(x = 0, 2, 4, 6) coated samples:

In this experiment, the matrix is quickly taken out after being placed in the film slurry. The Zr-V-Cr matrix samples were coated with four kinds of Zr_100-x_Cu_x_ (x = 0, 2, 4, 6) slurry with different Cu contents in Table 1. Sintering the prepared samples under 3.5 × 10^−3^ Pa, sintering at 900 °C for 60 min. (Zr_100-x_-Cu_x_)/(Zr-V-Cr)(x = 0, 2, 4, 6) samples were prepared.

(4)Test of sample inspiratory performance:

In this study, the sorption characteristics of getter was evaluated by constant volume method and constant pressure method. The National Standard (GB/T 25497-2010) [23] and American Society for Testing and Materials (ASTM f798-97 (2002)) [24] stipulate that CO and H_2_ can be used as test gases. In this study, the test gas of constant volume method was CO and the test gas of constant pressure method was H_2_. Both methods were completed in shanghai kingv material technology Co., Ltd. (Shanghai, China). The constant volume suction performance test system was JE-L1-02 and constant pressure method suction performance test system was JE-L1-01.

The constant volume method is to fill the test device with 40 Pa CO gas and record it as P_0_, recording the pressure value after 10 min as P_1_. The calculation method of the gettering capacity Q of the gettering material is as follows (1): m is the sample mass.
(1)$$Q=\frac{V\left(P_{0}-P_{1}\right)}{m}\;\left({\mathrm{cm}}^{3}\cdot s^{-1}\cdot g^{-1}\right)$$

The constant pressure method is to stabilize P_g_ at (4 ± 0.1) × 10^−4^ Pa, record the change values of P_m_ and P_g_, the test time is 2 h. The calculation method of the gettering rate S and the gettering capacity Q of the gettering material is as follows in (2) and (3):

F is the conductance value of the capillary (cm^3^·s^−1^); P_m_ is the pressure at the end of the capillary away from the aspirating material; P_g_ is the pressure at the end of the capillary near the getter material; t is inspiratory time; A is the surface area of the getter material (cm^2^) or mass (g).
(2)$$S=F\frac{p_{m}{\;-\;p}_{g}}{p_{g}\times A}$$
(3)$$Q=\frac{F}{A}\int_{0}^{t}{(p}_{m}{\;-\;p}_{g})\mathrm{dt}$$

(5)Microscopic analysis of samples and powders:

The laser particle size analysis of Zr-V-Cr matrix and Zr-Cu film powder was carried out, and the surface morphology of (Zr_100-x_-Cu_x_)/(Zr-V-Cr)(x = 0, 2, 4, 6) samples was observed by HITACHI SU1500.

(6)Test of mechanical properties and anti-dropping powders of samples:

In this study, the mechanical properties and anti-dropping powders of the samples were tested by a mechanical vibration test-bed, and the test methods and procedures of military standard (GJB 548b-2005) [25] microelectronic devices were used. The specific method is to put the sample into the experimental device, random vibration for 1H at 0–2000 Hz, root mean square value of acceleration 5.35 Grms. Then, the sample was detected by thermal infrared camera [26].

## 3. Results and Discussion

### 3.1. Effect of Cu Contents in Zr_100-x_Cu_x_ (x = 0, 2, 4, 6) Film on Sorption Performance of Coated Samples

The (Zr-Cu)/(Zr-V-Cr) samples with different Cu contents were sintered under 3.5 × 10^−3^ Pa, sintering at 900 °C for 60 min. The CO absorption curves of coated samples Zr/(Zr-V-Cr) sample, (Zr_98_Cu_2_)/(Zr-V-Cr) sample, (Zr_96_Cu_4_)/(Zr-V-Cr) sample, and (Zr_94_Cu_6_)/(Zr-V-Cr) sample were shown in Figure 2, and the performance results were shown in Table 2.

As seen in Figure 2 and Table 2, when Cu was not added to the film slurry, sample coated Zr film had the best sorption performance. With the increase of Cu content, CO sorption capacity and initial sorption rate gradually decreased. When the Cu content increased to 2 wt.%, the CO sorption capacity of (Zr_98_Cu_2_)/(Zr-V-Cr) sample decreased by 3.2% compared with Zr/(Zr-V-Cr) sample, When the Cu content increased to 4 wt.%, the CO sorption capacity of (Zr_96_Cu_4_)/(Zr-V-Cr) sample decreased by 11.6% compared with Zr/(Zr-V-Cr) sample. When the Cu content reached 6 wt.%, the CO sorption capacity of (Zr_94_Cu_6_)/(Zr-V-Cr) sample decreased significantly by 19.5% compared with Zr/(Zr-V-Cr) sample. It showed that when the Cu content was ≤2 wt.%, it had little effect on the CO sorption performance of the coated samples, but when the Cu content was ≥4 wt.%, the effect on the CO sorption performance of the coated samples gradually increased and the CO sorption performance decreased rapidly.

Since the sorption performance of the coated samples is closely related to the microstructure of films, the four kinds of coated samples with different Cu content (x = 0, 2, 4, 6) were characterized by SEM. The results were shown in Figure 3 to reveal the relationship between the sorption characteristics of the coated samples and the microstructure of Zr_100-x_Cu_x_ (x = 0, 2, 4, 6) films.

It was seen from Figure 3 that when the Cu content was 0 wt.% (Figure 3a), the surface of Zr/(Zr-V-Cr) sample contained a large number of unconnected particles, which had the risk of dropping powders from Zr-V-Cr matrix, so it could not be used. When the Cu content was 2 wt.% (Figure 3b), sintering neck was formed between the surface particles, and the surface porosity of (Zr_98_Cu_2_)/(Zr-V-Cr) sample did not decrease significantly compared with Zr/(Zr-V-Cr) sample, and the strength was higher than Zr/(Zr-V-Cr) sample. (Zr_98_Cu_2_)/(Zr-V-Cr) sample had the best microstructure. When the Cu content was 4 wt.% (Figure 3c), the sintered necks continued to grow, and the surface porosity continued to decrease. When the Cu content reached 6 wt.% (Figure 3d), the particles were completely melted and sintered together, so it had low surface porosity. This would prevent the active gases from entering the matrix and affect the sorption characteristics. With the decrease of porosity, the initial inspiratory rate and sorption capacity of CO also decreased.

In conclusion, when the Cu content was 2 wt.%, (Zr_98_Cu_2_)/(Zr-V-Cr) sample, which ensured the sorption characteristics, obtained the best microstructure, avoiding the risk of dropping powders from Zr-V-Cr matrix.

### 3.2. Study on the Sorption Performance of (Zr_98_Cu_2_)/(Zr_56.97_V_35.85_Cr_7.18_) Getter

Because the coated getter with Cu content of 2 wt.% obtained the best surface structure and the best getter performance, the following studies on the coated getter was (Zr_98_Cu_2_)/(Zr_56.97_V_35.85_Cr_7.18_) getter.

In order to further explore the influence of Zr_98_Cu_2_ gradient structure film on the gettering performance of coated samples, 3.2 further analyzed the influence of the film layer on the matrix from the CO gettering capacity, hydrogen absorption kinetic performance, and micro morphology of the matrix and film layer.

The sorption performance of getter was tested by two test methods: constant pressure and constant volume. The Zr-V-Cr matrix and (Zr_98_Cu_2_)/(Zr-V-Cr) getter were tested by constant volume method. The CO absorption curves are shown in Figure 4 and the data are shown in Table 3.

According to the data in Table 3 and Formula (1), the CO sorption capacities of matrix and (Zr_98_Cu_2_)/(Zr-V-Cr) getter at room temperature were 11,474 Pa·cm^3^·g^−1^ and 8827 Pa·cm^3^·g^−1^, respectively. Compared with the matrix samples, the CO pressure change value ∆P of the gradient structure (Zr_98_Cu_2_)/(Zr-V-Cr) getter decreased by 0.52 Pa, and the CO sorption capacity decreased by 2647 Pa·cm^3^·g^−1^, which decreased by 23% of the matrix. It was seen that Zr_98_Cu_2_ film has an impact on the CO sorption characteristics.

Based on the constant volume method, the constant pressure method was used to test the hydrogen absorption performance curves between Zr-V-Cr matrix, (Zr_98_Cu_2_)/(Zr-V-Cr) getter and SAES getter. It was seen from Figure 5 that Zr-V-Cr matrix had the best sorption characteristics, followed by gradient structure (Zr_98_Cu_2_)/(Zr-V-Cr) getter. SAES getter had the worst sorption characteristics.

The test data of constant pressure method are shown in Table 4. Compared with SAES getter, the initial hydrogen sorption rate of coated samples was increased by 43%, and the hydrogen absorption capacity of 1 h was increased by 93.5%. Although the sorption characteristics of coated samples was worse than Zr-V-Cr matrix, it was better than SAES getter. (Zr_98_Cu_2_)/(Zr-V-Cr) getter of this experiment could be used practically.

It can be seen from the inspiratory performance test that (Zr_98_Cu_2_)/(Zr-V-Cr) getter had good sorption characteristics. However, in order to judge whether it has a good anti- dropping powder performance, the laser particle size analysis of Zr_98_Cu_2_ powder and Zr-V-Cr matrix powder was carried out. The results were shown in Figure 6 and Table 5. It was seen that the average particle size of Zr-V-Cr matrix powder was 24.23 μm. Particle size of Zr_98_Cu_2_ slurry powder was 3.34 μm. The particle size of Zr_98_Cu_2_ slurry powder was smaller than Zr-V-Cr matrix powder. After powder metallurgy, the pore size of Zr_98_Cu_2_ film was smaller than Zr-V-Cr matrix as a whole, which effectively prevented the particles dropping powders from Zr-V-Cr matrix.

The performance of the samples was closely related to the microstructure—the microstructure of the surface and profile of the sample, respectively.

The SEM images of Zr-V-Cr matrix sample and (Zr_98_Cu_2_)/(Zr-V-Cr) getter after sintering are shown in Figure 7. The Zr-V-Cr matrix sample without Zr-Cu film had less sintering neck (Figure 7a), so there was a risk of dropping powders from Zr-V-Cr getter. When Zr_98_Cu_2_ film was coated on the Zr-V-Cr matrix, the sample produced high-density sintering neck after sintering (Figure 7b). The particles were in the state of fusion welding, which can effectively prevent the particles of the matrix from falling out through the pores of the film.

In order to explore the bond strength between Zr_98_Cu_2_ film and Zr-V-Cr matrix, the faultage image at the interface between film and matrix of (Zr_98_Cu_2_)/(Zr-V-Cr) getter was tested by SEM. The results are shown in Figure 8. It can be seen from Figure 8 that the surface of Zr-V-Cr matrix was covered with a layer of uniform pores with gradient structure, and the thickness is about 100 μm. At the interface between the film layer and the matrix, the particles of the Zr-V-Cr matrix and Zr_98_Cu_2_ film penetrated each other, which made the film layer closely combined with the Zr-V-Cr matrix and would not separate from the Zr-V-Cr matrix.

### 3.3. Analysis of Mechanical Properties of (Zr_98_Cu_2_)/(Zr-V-Cr) Getter

In order to further ensure the reliability of the getter during use, it is necessary to test the mechanical properties and the ability for anti-dropping powders of the getter, simulating its adaptation in the real environment or an even more severe environment. The (Zr_98_Cu_2_)/(Zr-V-Cr) getter was placed in a metal tube to test its mechanical strength under random vibration test. The control map of random vibration test is shown in Figure 9. The metal tube was sealed with a special transparent adhesive, and the infrared signal through the transparent adhesive was observed after random vibration to test whether the Zr_98_Cu_2_ film of the sample had fallen off.

Random vibration is the main cause of commodity damage during transportation. Random vibration can reflect the adaptability of the test piece to the vibration environment and assess its structural integrity. The random vibration simulation of this experiment is the random vibration in uncertain environment, which cannot be predicted and copied. It is closer to the real situation, and the results are more real and reliable.

Figure 9a reflects the control chart of Power Spectral Density (PSD) changing with time. The Power Spectral Density (PSD) is the vibration energy contained in each frequency unit. The green curve in the middle is the setting target curve, the upper two and the lower two are the protection curves (Orange) and termination curves (red) of the equipment, and the blue changing curve attached to the middle setting value is the actual control curve of the shaking table. The actual control curve will be within the protection curve in normal operation. Figure 9b reflects the time history of random vibration test and the feedback signal in the vibration process. The irregular fluctuation from the curve can indicate that the vibration is disordered and irregular, which conforms to the characteristics of random vibration and indicates that the random vibration of the machine is normal.

After random vibration test, no black particle dots were found in the infrared signal images of mechanical properties test results (Figure 10), which confirmed that the sample had no powder loss according to the results of thermal infrared detection. The random vibration test did not cause the (Zr_98_Cu_2_)/(Zr-V-Cr) getter to lose powders. There was no obvious deformation of the (Zr_98_Cu_2_)/(Zr-V-Cr) getter, the surface was intact without damage, and there was no powder loss. It satisfied the practical application conditions and could be applied to high vacuum fields such as hydrogen masers.

## 4. Conclusions

In view of the contradiction between strength and gettering performance and the poor anti-dropping powder’s ability of Zr_56.97_V_35.85_Cr_7.18_ in application, this paper mainly adopted the method of powder metallurgy, sintering the (Zr_98_Cu_2_)/(Zr_56.97_V_35.85_Cr_7.18_) getter under 3.5 × 10^−3^ Pa, and sintering at 900 °C for 60 min. The Zr_98_Cu_2_ film was closely combined with the Zr-V-Cr matrix. The particles on the surface of Zr-V-Cr getter can be prevented from falling off by the film layer, and the purpose of anti-dropping powders can be realized without reducing the getter performance.

The film layer was strengthened by adding element Cu into Zr powder, and the Zr-Cu alloy film had the properties of high strength, high hardness, corrosion resistance, and wear resistance. In the study, it was found that when the Cu content was 2 wt.%, (Zr_98_Cu_2_)/(Zr-V-Cr) sample, which ensured the sorption characteristics, obtained the best surface structure.The Zr-Cu gradient structure film with small pore size was formed by the Zr-Cu alloy powder with small particle size, which covered the surface of the Zr-V-Cr matrix with large particle size. So Zr-Cu film prevented the particles on the surface of the Zr-V-Cr alloy from falling off. It was proved that (Zr_98_Cu_2_)/(Zr-V-Cr) getter solved the problem of material powder dropping by SEM images, particle size analysis, and Infrared signal images.The getter performance experiment in this paper proved that the gettering performance of (Zr_98_Cu_2_)/(Zr-V-Cr) getter was superior to a product of SAES getters. Therefore, the coated sample had good gettering performance and met the use requirements. After random vibration test, the coated sample still had good mechanical properties. Therefore, (Zr_98_Cu_2_)/(Zr-V-Cr) getter solved the contradiction between strength and getter performance. It could be satisfactorily applied to the getter pump in hydrogen masers.

## Figures and Tables

**Figure 1 materials-15-06147-f001:**
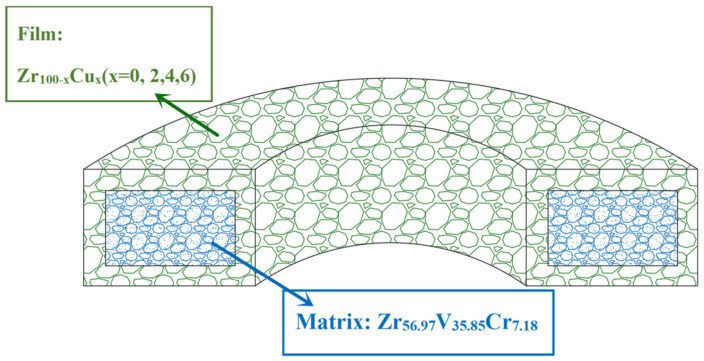
Design principal diagram of gradient structure Zr-V-Cr getter.

**Figure 2 materials-15-06147-f002:**
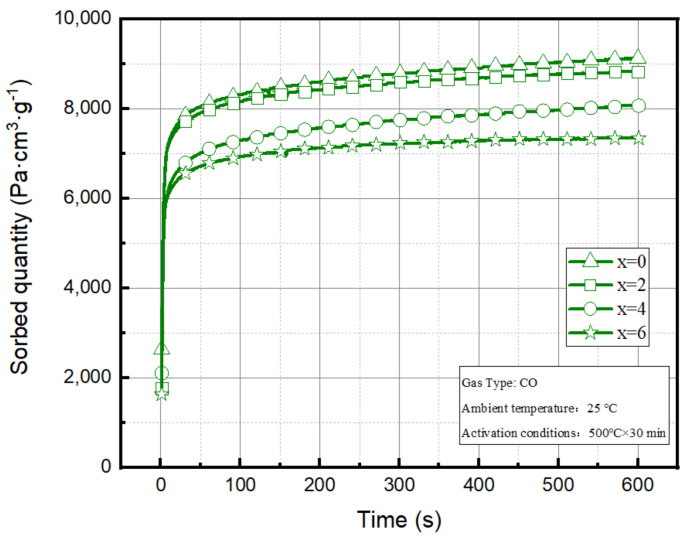
CO absorption curves of coated samples.

**Figure 3 materials-15-06147-f003:**
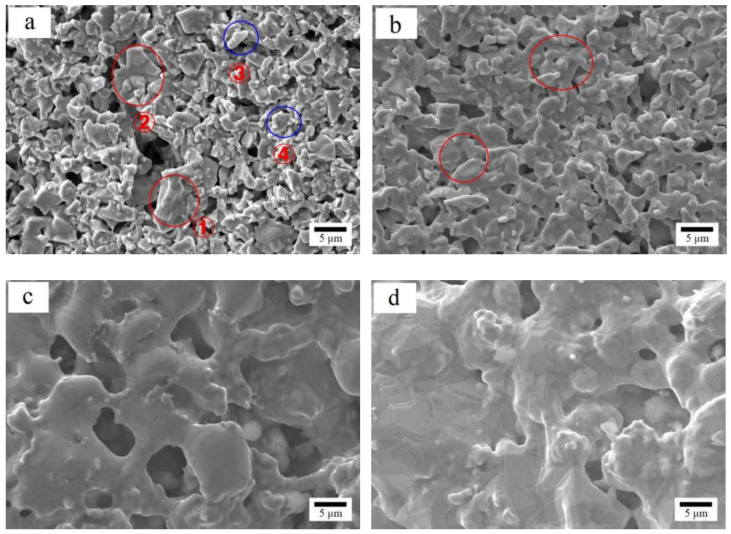
Microstructure of coated samples with different Cu contents by HITACHI SU1500 (2000×, Tokyo, Japan) (**a**) x = 0; (**b**) x = 2; (**c**) x = 4; (**d**) x = 6.

**Figure 4 materials-15-06147-f004:**
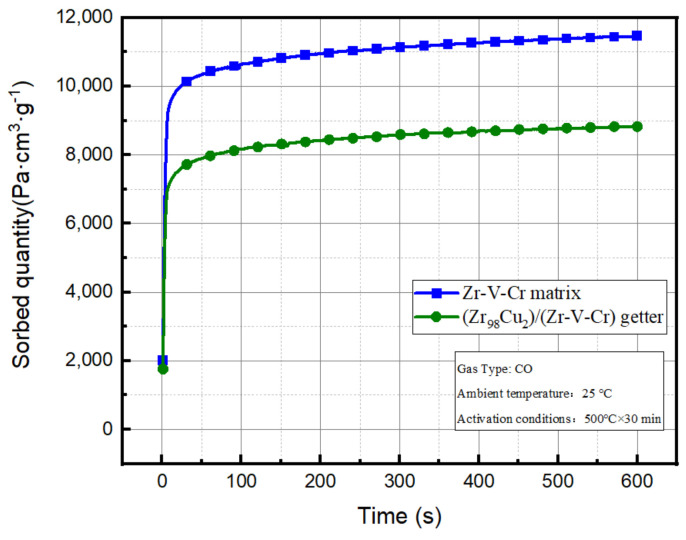
Comparison of CO absorption curves of Zr-V-Cr matrix and (Zr_98_Cu_2_)/(Zr-V-Cr) getter.

**Figure 5 materials-15-06147-f005:**
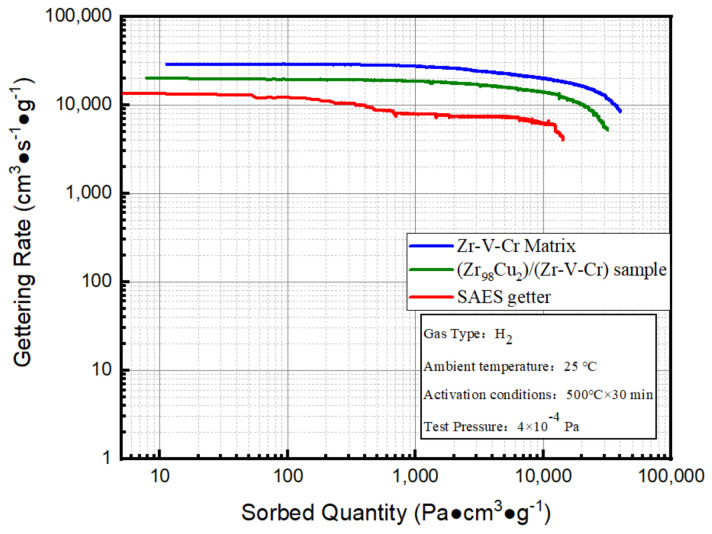
Comparison of hydrogen absorption performance between Zr-V-Cr matrix, (Zr_98_Cu_2_)/(Zr-V-Cr) getter, and SAES getter.

**Figure 6 materials-15-06147-f006:**
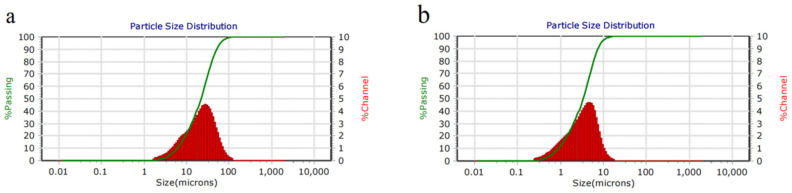
Laser particle size curves analysis of Zr-V-Cr matrix powder (**a**) and Zr_98_Cu_2_ slurry powder (**b**).

**Figure 7 materials-15-06147-f007:**
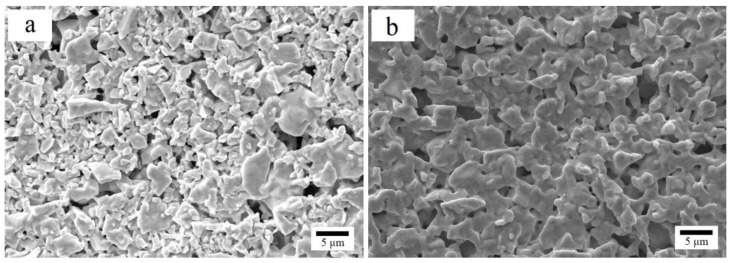
850 °C × 60 min substrate morphology (2000×) (**a**) and 900 °C × 60 min Zr_98_Cu_2_ film morphology (**b**).

**Figure 8 materials-15-06147-f008:**
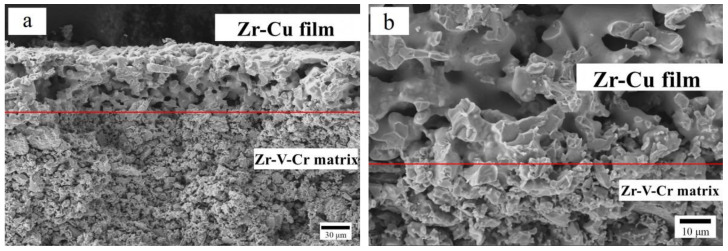
Faultage image at the interface of (Zr_98_Cu_2_)/(Zr-V-Cr) getter by HITACHI SU1500. (**a**) 300× SEM; (**b**) 1000× SEM.

**Figure 9 materials-15-06147-f009:**
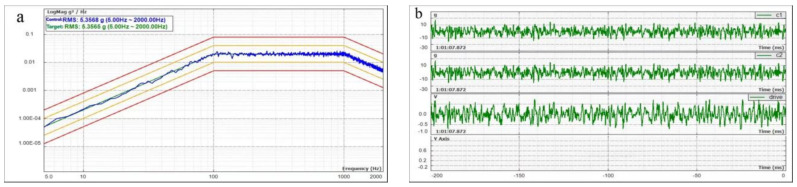
Frequency conversion vibration control spectrum (*X*-axis). (**a**) Control chart of power spectral density varying with frequency (**b**) Vibration time history.

**Figure 10 materials-15-06147-f010:**
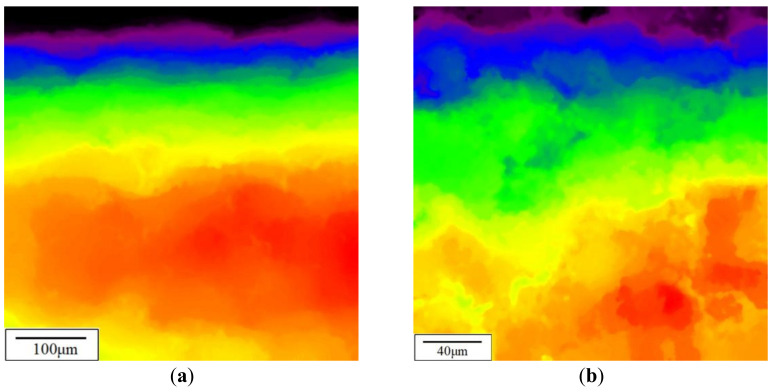
Infrared signal images after random vibration test. (**a**)100×; (**b**) 250× .

**Table 1 materials-15-06147-t001:** Composition of Zr_100-x_Cu_x_ (x = 0, 2, 4, 6) film.

X Value	Weight of Zr Powder (g)	Weight of Cu Powder (g)
0	15	0
2	14.7	0.3
4	14.4	0.6
6	14.1	0.9

**Table 2 materials-15-06147-t002:** CO sorption capacity parameters of coated samples.

Name	X Value	Initial Pressure P_0_ (Pa)	Extreme Pressure P_1_ (Pa)	Co Differential Pressure ∆P (Pa)	Co Sorption Capacity Q (Pa·cm^3^·g^−1^)	Performance Evaluation
Zr/(Zr-V-Cr) sample	0	40	21.781	18.219	9122	Good
(Zr_98_Cu_2_)/(Zr-V-Cr) sample	2	40	21.839	18.161	8827	Good
(Zr_96_Cu_4_)/(Zr-V-Cr) sample	4	40	21.989	18.011	8063	Poor
(Zr_94_Cu_6_)/(Zr-V-Cr) sample	6	40	22.134	17.864	7340	Poor

**Table 3 materials-15-06147-t003:** Comparison of CO sorption capacity parameters of Zr-V-Cr matrix and (Zr_98_Cu_2_)/(Zr-V-Cr) getter.

Sample Type	Initial Pressure P_0_ (Pa)	Extreme Pressure P_1_ (Pa)	Co Differential Pressure ∆P (Pa)	CO Sorption Capacity Q (Pa·cm^3^·g^−1^)	Performance Evaluation
Zr-V-Cr Matrix	40	21.319	18.681	11,474	Good
(Zr_98_Cu_2_)/(Zr-V-Cr) getter	40	21.839	18.161	8827	Good

**Table 4 materials-15-06147-t004:** Comparison of hydrogen absorption performance of Zr-V-Cr matrix, (Zr_98_Cu_2_)/(Zr-V-Cr) getter, and SAES getter.

Sample Type	Initial Hydrogen Absorption Rate (cm^3^·s^−1^·g^−1^)	1 h Hydrogen Absorption Capacity (Pa·cm^3^·g^−1^)	2 h Hydrogen Absorption Capacity (Pa·cm^3^·g^−1^)
Zr-V-Cr Matrix	28,822	25,298	40,017
(Zr_98_Cu_2_)/(Zr-V-Cr) getter	20,061	20,168	31,107
SAES getter	13,435	10,422	/

**Table 5 materials-15-06147-t005:** Particle size parameters of Zr-V-Cr matrix powder and Zr_98_Cu_2_ slurry powder.

Sample Type	Maximum Particle Size (μm)	Minimum Particle Size (μm)	Average Particle Size (μm)	≥10 μm	≥5 μm
Zr-V-Cr matrix powder	124.5	1.783	24.23	78%	93%
Zr_98_Cu_2_ film slurry powder	18.5	0.265	3.34	4%	30%

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
