# Peer review of "Study on Coated Zr-V-Cr Getter with Pore Gradient Structure for Hydrogen Masers"

_materials, 2022, doi:10.3390/ma15176147_

Round 1

Reviewer 1 Report

I would like to recommend publishing this work after addressing the following points:

1. Introduction is well-organized and well-written, but the importance and novelty of the research should be highlighted and more clearly stated. The authors give some examples of works in the bibliography, but which is the advantage of their work in comparison with those works.

2. In Materials and method section, please provide the purity of your chosen precursors.

3. In Materials and method section, the conditions used for all characterization techniques should be added.

4. Author Should provide more information about the preparation technique, in which the procedure of the method used is not unclear.

5. The authors are responsible for the English, which should be polished throughout the manuscript to clear some minor typo/grammar errors.

4. In the introduction part, Some publications are suggested to refer to improve the quality of the manuscript, such as: https://doi.org/10.1016/j.jtice.2021.08.034, https://doi.org/10.1007/s10562-022-04026-y, DOI: 10.1039/D0RA09970H.

5. The author should better improve the beauty and quality of the figures in the manuscript.

Reviewer 2 Report

In this work, the Zr-Cu film is coated on the Zr-V-Cr matrix, and then sintered to the gradient change of pore structure from the matrix to the film. The film is firmly bonded to the matrix, so the getter has good mechanical properties and anti-dropping powders ability while maintaining high sorption characteristics. The structural schematic and the mechanical properties of microelectronic devices of (Zr-Cu)/(Zr-V-Cr) getter is tested to ensure practical application in the hydrogen maser. I appreciate the efforts of the author, but the materials are not deeply analyzed and the presentation skill is also very weak. So, to help the author to reach the standard of materials-MDPI, there are some comments.

1. The Abstract part is too lengthy (250 words). Can the authors compact it to 200 words to make it concise?

2. In some figures, Chinese/Korean is written. This must be removed or converted to English language.

3. FTIR / Raman results are missing. Please provide them too in order to confirm the synthesis of desired materials with associated functional groups.

4. In section 2. Materials and Methods, please provide a list of materials in paragraph form along with their origin and purity. For example: Chemical Engineering Journal 441, 136063 & Chemical Engineering Journal 420, 130529.

5. The model number and specifications of SEM machine are not provided in the manuscript. Please provide them too. For example: Nanomaterials 11 (11), 2908 & Journal of Industrial and Engineering Chemistry 98, 283-288.

6. There is no connection in the result and discussion section. I strongly recommend the author please connect the paragraphs of result and discussion with each other in a logical way, like "In order to explore the morphology of synthesized material" or "to shed more light on the synthesized material," etc.

7. In figure 5. Comparison of hydrogen absorption performance between Zr-V-Cr matrix, (Zr98Cu2)/(Zr-V-Cr) getter and SAES getter. The curve of “SAES getter” is comparatively noisy. Can the authors re-do this measurement?

8. This work should be compared with recently published at least 8 articles in a scientific table.

9. Make sure all abbreviations are written out in full the first time used. This is particularly important in the abstract and the conclusions but work through the entire ms carefully from this perspective.

10. There are only analyzed through SEM results, Pore Gradient Structure, Random vibration test and Thermal infrared detection in the manuscript. So, the author should also provide some other results related to the sample characterization XRD, XPS, TEM, Elemental mapping, HRTEM, etc. Or justify that why they are not required?

11. Please improve the last paragraph of introduction, figures must not be referred in this paragraph, else a concise outline of work to be presented herein.

12. Manuscripts published in materials-MDPI must explain the significant advances provided in approaches and understanding compared to previous literature, and/or demonstrate convincingly potential in new applications. The Conclusions of your paper are especially important for this. Therefore, please try to sharpen this further. The optimal Conclusion should include:

* A summary of your key findings.

* A highlight of your hypothesis, new concepts, and innovations.

* A summary of key improvements compared to findings in the literature [provide a couple of references to indicate key improvements].

* Your vision for future work.
